# Binding and Neutralizing Capacity of Respiratory Syncytial Virus (RSV)-Specific Recombinant IgG Against RSV in Human Milk, Gastric and Intestinal Fluids from Infants

**DOI:** 10.3390/nu12071904

**Published:** 2020-06-27

**Authors:** Veronique Demers-Mathieu, Jiraporn Lueangsakulthai, Yunyao Qu, Brian P. Scottoline, David C. Dallas

**Affiliations:** 1Nutrition Program, School of Biological and Population Health Sciences, College of Public Health and Human Sciences, Oregon State University, Corvallis, OR 97331, USA; lueangsj@oregonstate.edu (J.L.); yunyao.qu@oregonstate.edu (Y.Q.); Dave.Dallas@oregonstate.edu (D.C.D.); 2Department of Pediatrics, Oregon Health & Science University, Portland, OR 97239, USA; scottoli@ohsu.edu

**Keywords:** passive immunity, neonatal infection, pathogen-specific antibody

## Abstract

Oral administration of pathogen-specific recombinant antibodies may help to prevent infant gastrointestinal (GI) pathogen infection; however, to neutralize an infectious agent, these antibodies must resist degradation in the GI tract. Palivizumab, a recombinant antibody specific for the respiratory syncytial virus (RSV), was used as a model for pathogen-specific IgG in human milk. The aim was to compare the remaining binding capacity of palivizumab in milk between three mothers after exposure to an in vitro model of infant gastrointestinal digestion (gastric and duodenal fluids) using ELISA. The neutralizing capacity of palivizumab in pooled human milk, gastric contents, and stools from preterm infants was also evaluated for blocking RSV with green fluorescent protein (RSV-GFP) infection in Hep-2 cells using confocal and inverted microscopy and flow cytometry. The reduction of palivizumab binding capacity in human milk and digested samples was slightly different between mothers. Overall, palivizumab decreased 50% after simulated gastric digestion with pepsin and 62% after simulated intestinal digestion with pancreatin. Palivizumab (2–8 μg/mL) in human milk or stool samples blocked RSV (3.4 × 10^4^ FFU/mL) infection (no syncytia formation on Hep-2 cells) by microscopy. Syncytia formation was detected on Hep-2 cells when RSV was incubated in gastric contents or virus medium with 2–4 μg/mL of palivizumab, but no infection was observed at 8 μg/mL. No fluorescence (absence of infected cells) was detected when palivizumab (100 μg/mL) was incubated in human milk or medium with RSV-GFP (1.1 × 10^5^ FFU/mL), whereas fluorescence increased with the reduced concentration of palivizumab using flow cytometry. These results suggest that undigested and digested matrices could change the binding and neutralizing capacity of viral pathogen-specific antibodies.

## 1. Introduction

Respiratory syncytial virus (RSV) can induce severe infections in preterm infants, such as pneumonia or bronchiolitis [1]. This viral infection can lead to serious complications, re-hospitalization, and death. The Institute of Allergy and Infectious Diseases reported that 75,000 to 125,000 infants in the United States are hospitalized for RSV infections each year. Human milk contains immunoglobulins that are RSV-specific and may help prevent RSV infection [2]. Secretory IgA (SIgA) is the most abundant antibody in human milk (~80%) and other mucosal secretions in the human body, and its main role is to block the attachment and/or neutralize infectious pathogens on the surface of intestinal epithelial cells [3]. On the other hand, immunoglobulin G (IgG) is the most abundant antibody in the blood and efficiently interacts with the immune cells to activate the immune responses (opsonization) when pathogens have entered the bloodstream [4]. While human milk SIgA is likely to stay on the surface of the intestinal mucosa, human milk IgG may potentially be absorbed in the gut to reach the bloodstream, enabling interaction with other immune components to protect against systemic pathogen invasion. However, no study has demonstrated the absorption of human milk IgG across the human gut into the bloodstream. After entering the blood, RSV-specific IgG could modulate the adaptive immune system to prevent RSV infection in the respiratory tract. More study is needed to determine whether human milk IgG can be absorbed by the gut and enter the circulatory system in humans. As IgG concentration in human milk is low (~4% of total Ig at 1–3 weeks of postpartum time) compared with SIgA/IgA (~80%) and SIgM/IgM (~16%) [5,6], it is possible that recombinant IgG supplementation could increase the capacity to prevent RSV infection, if indeed, these antibodies can be absorbed. 

In contrast to respiratory pathogen prevention with oral antibodies, which may require systemic absorption, enteric pathogen prevention with orally-administered antibodies would require functional survival across the proteolytically active GI tract to prevent adhesion of pathogens to the intestinal epithelium. Whether orally-administered recombinant antibodies survive across the infant digestive tract has not been examined. To begin to understand the potential for using recombinant oral antibodies for infection prevention, we selected a model recombinant humanized IgG1 monoclonal antibody, palivizumab, which is Federal Drug Administration (FDA)-approved to prevent respiratory syncytial virus (RSV) infections via intramuscular injection in high-risk infants [7]. Palivizumab binds to an epitope in the A antigenic region of RSV F protein. In conjunction with obtaining “may proceed” approval from the FDA to use palivizumab enterally in infants, we examined the degradation of palivizumab exposed to simulated gastrointestinal conditions. Our related work will examine the in vivo survival of palivizumab in infants when administered enterally.

Relatively few studies have examined the digestion of antibodies. Eibl et al. [8] demonstrated that a fraction of a large dose (~126 mg) of purified IgG from human serum (as a supplement to the infant feeding of formula or formula plus pasteurized, pooled human milk) survived intact to the stool of preterm infants. Our recent studies demonstrated that human milk IgG concentration decreased 77% in the term stomach at 2 h post-ingestion [5] but was stable in the preterm stomach [5,9]. This difference was likely due to the higher gastric digestion capacity for human milk protein in term infants compared with preterm infants [10].

Our recent study demonstrated that palivizumab (IgG, sIgA and IgA formats) is degraded during incubation in infant gastric and intestinal fluids (ex vivo digestion) [11]. In the present study, we assessed the binding and neutralizing capacity of palivizumab (IgG format) supplemented in human milk across in vitro infant digestion. The in vitro infant digestion model eliminates the variability of gastric and duodenal fluid digestive activity between infants; with the limitation of not reflecting the breadth of biological variability in proteolytic degradation. As binding capacity is associated with neutralizing capacity [12], a loss of binding capacity for palivizumab across simulated digestion would indicate functional degradation. As a complement to the simulated digestion experiments, the neutralizing capacity of palivizumab was evaluated in human milk, gastric contents, and stool samples from preterm infants (ex vivo) mixed with RSV and incubated on Hep-2 cells and assayed using confocal and inverted microscopy as well as flow cytometry. This study provides an important step to determining the minimal amount of model recombinant IgG needed in human milk to obtain quantitation of surviving and active palivizumab in gastric and intestinal contents from infants after feeding for future clinical study.

## 2. Materials and Methods

### 2.1. Human Milk Collection

Human milk (raw and non-pasteurized) was donated by the Northwest Mothers Milk Bank (Beaverton, OR, USA) from individual sources and pools. Mothers who donated to the mother’s milk bank signed a consent to use their milk for research. This study was approved by the Institutional Review Board of Oregon Health and Sciences University (STUDY 00018274). Milk samples were collected at home with clean electric breast pumps into sterile plastic containers and stored immediately in −20 °C freezers. After donation, milk samples were stored at −20 °C at the Northwest Mothers Milk Bank, transported on dry ice to Oregon State University, and then stored at −80 °C until use.

### 2.2. Effect of Heating, Freeze-Thaw Cycles, and Incubation on the Stability of Palivizumab in Human Milk

A final concentration of 100 μg/mL of palivizumab in human milk feeding was selected to obtain a higher level of IgG than that present in human milk (~20 μg/mL) in order to determine the remaining binding capacity at the end of the digestion above that of the endogenous antibody. To examine the effects of heating and freeze-thaw cycles on palivizumab binding, a pool of human milk samples was created. Palivizumab (Synagis^®^, MedImmune, Gaithersburg, MD, USA) was added to 4 mL of pooled human milk and mixed by pipetting, which was divided into 200 μL aliquots. The following treatments were performed in triplicate: Incubation on ice (control); incubation at 37 °C for 1 h with mixing at 300 rpm; heating at 55 °C for 30 min (a treatment often used to inactivate complement); heating at 100 °C for 30 min to evaluate for the total inactivation of palivizumab in human milk, freeze/thaw cycling via freezing at –20 °C for 1 h followed by thawing at 37 °C for one min, and freezing at –80 °C for 1 h, followed by thawing at 37 °C for 1 min. Palivizumab was also incubated at 37 °C for 2 h with mixing at 300 rpm and measured every 30 min in the pool of human milk samples. Samples were stored on ice for maximum 30 min following each treatment before assaying for palivizumab binding capacity. Human milk without palivizumab was used as a blank for the standard curve.

### 2.3. Effect of pH on Palivizumab Stability

To determine the effect of pH on the binding capacity, palivizumab was added at 100 μg/mL in milk adjusted to pH 4 (simulating the acidic pH in infant postprandial gastric fluids), pH 7 (simulating the neutral pH in human milk) or 8 (simulating the alkaline pH in duodenal fluid). Binding capacity of palivizumab in these mixtures was measured before and after 2 h of incubation at 37 °C with shaking at 300 rpm. Human milk was added to the dilution buffer for the standard curve.

### 2.4. Palivizumab Stability in Simulated Gastrointestinal Fluids

Palivizumab was added in human donor milk from three different mothers (collected during the first week of lactation) and incubated in gastric and duodenal fluids (in vitro) to simulate gastrointestinal digestion in infants. The in vitro infant digestion model was adapted from previous publications [13,14]. Palivizumab was added into 1 mL of human milk at 100 μg/mL to perform in vitro gastric and intestinal digestion. Simulated gastric fluid prior to addition of protease was 3.0 mM NaCl, 14.7 mM KCl, 1.0 mM CaCl2, and 18 mM HCl (all products from Sigma-Aldrich, St. Louis, MO, USA) in nanopure water at pH 4.0 adjusted by adding 1M HCl dropwise while mixing. Simulated duodenal fluid without protease was 0.15 M NaCl, 2 mM bile salts (50% cholic acid sodium, 50% deoxycholic acid) (all products from Sigma-Aldrich) in nanopure water) at pH 8.0 adjusted by adding 1M NaOH dropwise while mixing. The experimental conditions were donor human milk no treatment control; incubation at 37 °C for 1 h with shaking at 300 rpm in simulated gastric fluid; incubation at 37 °C for 1 h with shaking at 300 rpm in simulated gastric fluid with 22.75 U pepsin/mg protein (based on the assumption that human milk is 10 mg/mL protein) (Sigma-Aldrich); incubation for 1 h with shaking at 300 rpm in simulated gastric fluid followed by 2 h with shaking at 300 rpm in simulated duodenal fluid without protease; and incubation at 37 °C in simulated gastric fluid with pepsin for 1 h with shaking at 300 rpm, followed by incubation at 37 °C for 2 h with shaking at 300 rpm in simulated duodenal fluid with 3.5 U porcine pancreatin/mg protein (Sigma-Aldrich). Simulated fluid was added to the dilution buffer for the standard curve to compensate the potential background effects of simulated fluid during ELISA.

### 2.5. Palivizumab Stability in Simulated Duodenal Fluids in the Presence of Bile Salts

To determine the potential interaction between bile salts on palivizumab binding in the simulated intestinal digestion fluid without the contribution of pancreatin, palivizumab was added at 100 μg/mL in the following mixtures: donor human milk with no treatment (control), PBS pH 8.0, human milk in PBS, bile salts in PBS, and human milk with bile salts in PBS. The pH of the bile salt-containing reaction was adjusted to pH 8.0 to simulate the pH of intestine, and its concentration was prepared as described in the previous paragraph. All reactions with bile salts were conducted without pancreatin to allow detection of the impact of bile salts on palivizumab binding capacity without added protease. Reactions were performed at 37 °C for 2 h with shaking at 300 rpm. Binding capacity of palivizumab in these mixtures was measured before and after incubation. Milk, bile, or PBS were used as blanks to create three standard curves.

### 2.6. Palivizumab ELISA

The spectrophotometric ELISAs were recorded with a microplate reader (Spectramax M2, Molecular Devices, Sunnyvale, CA, USA) with three replicates of blanks, standards, and samples. All ELISAs were performed according to the methods described by Bio-Rad with some modifications. This method provides a procedure for carrying out a Palivizumab ELISA with human anti-palivizumab antibodies, which are paratope specific, high affinity, anti-idiotypic antibody that specifically recognizes the monoclonal palivizumab IgG (Figure 1). Briefly, 100 μL per well of 1 μg/mL anti-palivizumab idiotype antibody HCA261 (Bio-Rad, Richmond, CA, USA) in 1× PBS was coated onto a clear flat-bottom Maxisorp 96-well plate (Nunc, Thermo Scientific, Waltham, MA, USA) and incubated overnight at 4 °C. After incubation, the microtiter plate was washed three times with PBS with 0.05% Tween-20 (PBST) (Bio-Rad) and blocked for 1 h with 100 μL of PBST with 5% bovine serum albumin (BSA) (Blocker^TM^ BSA (10×) in PBS, Thermo Scientific) at room temperature (RT). After washing three times, standards and samples were added to the wells (100 μL) and incubated for 1 h at RT. Standards were prepared using palivizumab in serial dilutions (from 1–10,000 ng/mL) in PBST with 10% human AB serum (Corning, Manassas, VA, USA) at 1:250 of the specific fluid used (milk) to compensate for any background effect of the matrices. Fluid samples were diluted 50× and 200× (data were averaged) with PBST supplemented with 10% of human AB serum, added in wells (100 μL) and incubated at RT for 1 h. After incubating and washing, horseradish peroxidase (HRP)-conjugated anti-palivizumab detection antibody HCA262P (Bio-Rad) was diluted in HISPEC immunoassay diluent (Bio-Rad) at 2 μg/mL, added to wells (100 μL) and incubated at RT for 1 h. After the plates were washed three times with PBST, 100 μL of the substrate 3,3′,5,5′-tetramethylbenzidine (1×, Invitrogen, San Diego, CA, USA) was added to the wells and incubated for 5 min at RT followed by addition of 50 μL of 2N sulfuric acid to stop the color reaction. Optical density was measured at 450 nm.

### 2.7. Individual Mother’s Milk, Effect of Bile Salts, and pH

ELISAs were performed as specified for pooled donor milk with the following differences. After adding the capture antibody and washing, the wells were blocked for 1 h with 150 μL of PBST with 1% of BSA at RT. A range of 0 to 1000 ng/mL palivizumab in PBST with 1% BSA with no sample background added was used for the standard curve. All samples were diluted 200× and 400× in 1% BSA in PBST. For detection, 0.16 μg/mL goat anti-human IgG-HRP was used. The results from the 200× and 400× dilutions were averaged.

### 2.8. Neutralizing Capacity of Palivizumab against Human Respiratory Syncytial Virus with Green Fluorescent Protein (RSV-GFP)

#### 2.8.1. Preparation of RSV-GFP Frozen Stock

HEp-2 cells (ATCC^®^ CCL23™; up to 5 passages, American Type Culture Collection, Manassas, VA, USA) were seeded in a tissue culture flask (75 cm^2^) with cell medium (DMEM containing 10% fetal bovine serum (FBS) and 1% antibacterial-antimycotic solution) and allowed to grow until reaching > 95% confluency (typically 24–48 h) in a tissue culture incubator at 37°C, 5% CO_2_. All media and solutions used for cell culture and virus media were from Life Technologies (Carlsbad, CA, USA). The cell monolayer was washed three times with sterile Hank’s balanced salt solution (HBSS) and infected with 1 mL of frozen RSV-GFP stock (ViraTree, Research Triangle Park, NC, USA) (RSV subtype A2 where GFP gene was inserted 5′ of NS1 gene) in 3 mL of virus medium (DMEM with antibiotics-antimycotics without serum). The flask was incubated at 37 °C, 5% CO_2_ in the tissue culture incubator for 2 h and rocked North-South and East-West directions every 15 min. After 2 h, 10 mL of the virus medium was added to stop virus adsorption. The flask was examined every day during incubation via an inverted microscope for cytopathic effects (CPE), namely syncytia formation, rounding, and sloughing, to ensure the viral infection had taken place. After 5 days post-infection, the spent media was forcefully pipette mixed 10 times to free the infected, weakly attached cell monolayer from the flask and collected in a 50 mL falcon tube. The pooled cells and supernatants were centrifuged at 280 g, 4 °C for 5 min and the supernatant was collected, leaving approximately 200 μL of supernatant in the tube with the pelleted cells. The cell pellet was resuspended with the leftover 200 μL of supernatant and frozen immediately in dry ice, followed by quickly thawing in a 37 °C water bath. This freezing/thawing step was repeated 3 times and vortexed after each cycle. All the freeze-thawed cell debris was pooled with the saved supernatant, and sterile glycerol was added at 15% (v/v) and mixed with a vortex device. The virus suspension was aliquoted into cryovials (300 µL/cryovial) and stored at −80 °C for long-term storage.

#### 2.8.2. Preparation of Human Milk, Gastric and Stool Samples from Preterm Infants

Samples were collected from six premature-delivering mother-infant pairs ranging in gestational age from 26–29 weeks in the NICU at the Randall Children’s Hospital at Legacy Emanuel NICU. This sample collection was approved by the Institutional Review Boards of Legacy Health Systems and Oregon State University. Eligibility criteria included having an indwelling naso/orogastric feeding tube and bolus feeding (<60 min infusion tolerated). Exclusion criteria included neonates with diagnoses that were incompatible with life, gastrointestinal system anomalies, major gastrointestinal surgery, severe genitourinary anomalies, and significant metabolic or endocrine diseases. Human milk and gastric samples (1–2 mL) were collected on days 8–9 of life. Human milk was fed to the infant via a nasogastric tube with a feeding pump set to deliver the entire bolus over 30–60 min. A 2-mL sample of the gastric fluid was collected 30 min after the completion of feed infusion. Stool (1 g) was collected within 48 h of the gastric sampling time point and was recovered from the diaper and scraped into a sterile jar. After collection of each sample type (human milk, gastric and stool), samples were placed immediately on ice and stored at −80 °C in the NICU. Samples were transported on dry ice to Oregon State University for sample analysis.

Human milk and gastric samples were rapidly thawed at 37 °C and centrifuged at 1301 × g for 20 min at 4 °C. The six human milk infranate samples were collected, pooled together, and separated into aliquots (200 μL). This same procedure was accomplished for the six gastric content samples. Frozen stool samples (0.2 g) were diluted in 1 mL of phosphate-buffered saline pH 7.4 (Thermo Fisher Scientific, Chino, CA, USA), mixed by vortex for 2 min, and then centrifuged at 1301× *g* for 20 min at 4 °C. The six supernatants were collected, pooled together, and separated into aliquots (200 μL). Pooled human milk, pooled gastric contents, and diluted stools were diluted 1:10, 1:20, and 1:25, respectively, with virus medium and centrifuged at 1301× *g* for 20 min at 4 °C. The supernatant was collected by pipette from below the upper-fat layer and filtered through syringe filters (GE Healthcare Whatman Uniflo Filters, 0.22 μm, PES filter media, Thermo Fisher Scientific) under sterile conditions, aliquoted into sterile vials, and stored at −80 °C until use in the neutralization assay.

#### 2.8.3. Neutralizing Capacity by Microscopy

HEp-2 cells were seeded onto a 48-well plate at a density of 3.5 × 105 cells/mL in cell medium and grown at 37 °C, 5% CO2, until the cells reached 80–90% confluency. The neutralizing capacity of palivizumab against RSV-GFP was examined by adding 100 μL of virus medium or diluted biological samples (supernatant of human milk, gastric contents or stools filtered (0.22 μm), containing 0, 4, 8 or 16 μg/mL of palivizumab). RSV-GFP suspension (100 μL) containing 50% tissue culture infectious dose (TCID50: 3.4 × 10^4^ focus forming units (FFU)/mL) was added in each vial, except one vial per type of sample (negative control) for 2 h at 37 °C, 5% CO_2_, and gently mixed every 30 min. Subsequently, the mixtures were transferred (200 μL) to confluent Hep-2 in the 48-well plate, incubated for 2 h at 37 °C. After the incubation, the mixtures were aspirated, and the cells were washed once with virus medium. Virus medium was added (200 μL) and cultured for 3 days at 37 °C. After infection, media was aspirated, cells were fixed with 4% paraformaldehyde (Thermo Fisher Scientific) and then washed with PBS. A blue fluorescent nucleic acid stain, 4,6-diamidino-2-phynylindole dilactate (DAPI, BioLegend, San Diego, CA, USA) was added at 300 nM for 5 min to allow immunofluorescence staining of Hep-2. GFP expression in infected cells was monitored using a confocal microscope system (Zeiss LSM 780 NLO, White Plains, NY, USA), which allowed for the rapid assessment of the presence of infected GFP-expressing cells in the monolayer. The neutralizing capacity of palivizumab in each condition was also monitored using an Olympus IX51 fitted with a Q Image camera (Olympus, Waltham, MS, USA) to compare the CPE. Three replicates were performed for each condition.

#### 2.8.4. Neutralizing Capacity by Flow Cytometry

HEp-2 cells were seeded onto a 48-well plate at a density of 3.5 × 10^5^ cells/ml in cell medium and grown at 37 °C, 5% CO_2,_ until the cells reached 80–90% confluency. The neutralizing capacity of palivizumab against RSV-GFP was performed by adding 100 μL of virus medium or pooled human milk (skimmed and filtered (0.22 μm) containing 0, 6.25, 25, 50, 100 μg/mL of palivizumab. RSV-GFP suspension (100 μL) containing 1.1 x 105 FFU/mL was added in each vial, except one vial per type of sample (negative control) for 2 h at 37 °C, 5% CO2, and gently mixed every 30 min. Subsequently, the mixtures were transferred (200 μL) to confluent Hep-2 in the 48-well plate, incubated for 2 h at 37 °C. After the incubation, the mixtures were aspirated, and the cells were washed once with virus medium. HEp-2 cells were resuspended in flow cytometry buffer (PBS, 2% FBS, 1 mM EDTA), washed twice with flow cytometry buffer, and resuspended in the same buffer. Cells were stored in PBS with Prolong diamond antifade molecular probes (Life Technologies) to maintain GFP fluorescence and analyzed after cell harvest for the relative fluorescent intensity of GFP. Data were acquired using FACSCalibur (BD Biosciences, San Jose, CA, USA). Data were analyzed using Summit software (DakoCytomation, Fort Collins, CO, USA). Three replicates were performed for each condition.

### 2.9. Statistical Analysis

One-way ANOVA followed by Dunnett’s multiple comparison test was used to compare palivizumab binding capacity within the incubation, and heat and freeze-thawing experiments. One-way ANOVA followed by Tukey’s multiple comparison test were applied to compare the palivizumab concentration detected by ELISA in the samples across in vitro digestion as well as the fluorescence detected by flow cytometry in Hep-2 cells with RSV-GFP and different palivizumab concentrations. Unpaired t-tests were used to compare the reduction of palivizumab in human milk between mother’s milk 2 and mother’s milk 1 or 3. Two-way ANOVA followed by Tukey’s multiple comparison test were applied to compare samples with and without bile salts and with different pH (GraphPad Prism software, version 8.2.1, San Diego, CA, USA). Differences were designated significant at *p* < 0.05.

## 3. Results

### 3.1. The Effect of Incubation, Heat, and Freeze-Thawing on Palivizumab

Palivizumab binding capacity was reduced 22.5% when human donor milk was incubated at 37 °C for 1 h (*p* = 0.002, Figure 2A). Palivizumab was stable when human donor milk was heat treated at 55 °C for 30 min but was completely degraded (100% loss, not detectable) after heat treatment at 100°C for 30 min (*p* < 0.001). Palivizumab was stable after freezing and thawing at –20 °C or –80 °C (Figure 2A). Palivizumab in human milk incubated at 37 °C reduced 19.1% after 30 min (*p* = 0.026), 19.8% after 60 min (*p* = 0.022), and ~24–25% (*p* = 0.004) after 90 and 120 min (Figure 2B). 

### 3.2. In Vitro Digestion of Palivizumab in Three Mother’s Milk Samples and a Pooled Pasteurized Donor Milk Sample with and without Proteases

Palivizumab in milk from three different mothers incubated with simulated term infant gastric fluids for 60 min (pH 4.0) was not stable either with or without pepsin. In each individual mother’s milk (Appendix A), recombinant IgG decreased during incubation in simulated gastric and intestinal contents. The reduction of palivizumab in human milk from mother 2 was higher than mother 1 (*p* = 0.049) or mother 3 (*p* = 0.004).

With the results of each mother’s milk averaged, palivizumab decreased 50% with pepsin and 40% without pepsin in gastric fluids (Figure 3). After simulated gastric digestion, palivizumab in human milk incubated 2 h with simulated term infant intestinal fluids (pH 8.0) was not stable either with or without pancreatin (Figure 3). Palivizumab decreased 62% compared with the original milk in the presence of pancreatic proteases (trypsin, chymotrypsin and elastase) and 46% in simulated intestinal fluid without proteases (Figure 3). Palivizumab decreased significantly more in the presence of pancreatic proteases than without proteases (Figure 3). 

### 3.3. The Effect of Bile Salts and pH on Palivizumab

Palivizumab concentration in milk after 2 h incubation with bile salts (pH adjusted to 8.0 without protease) was not significantly decreased. The reduction of palivizumab from time 0 to 2 h did not significantly differ between milk samples treated with bile salts and milk with no treatment (HM), suggesting that bile salts do not affect the reduction of palivizumab (Figure 4A). Adjusting milk to pH 4, 7, and 8 did not affect palivizumab stability (Figure 4B).

Appendix A shows the standard curves with milk as background at different concentrations. The addition of milk at 0.04%, 0.4%, or 4% was not significantly different from background without milk.

### 3.4. The Neutralizing Capacity of Palivizumab on RSV-GFP

With the confocal microscope, no positive cells (infected/expressed GFP cells) were observed when palivizumab (8 μg/mL) was incubated in pooled human milk with RSV-GFP (3.4 × 10^4^ TCID50/mL), suggesting that palivizumab blocked the infection of Hep-2 cells (Figure 5). In the absence of palivizumab, the Hep-2 cells expressed GFP when incubated with RSV and pooled human milk, indicating the presence of infected cells with RSV (Figure 5). In the absence of RSV, palivizumab in pooled human milk did not affect the monolayer cells and no GFP expression was observed.

With the inverted microscope, no syncytia formation was observed with palivizumab (8, 4, and 2 μg/mL) in pooled human milk or in infant stools whereas some syncytia formation (large black spots) were detected in gastric contents or medium with palivizumab (Figure 6F–J,Q–U). These results indicate that palivizumab blocked more RSV infection in milk or stool than in medium or gastric contents. It also could be related to a more inherent viral neutralizing capacity in human milk or stool matrix than in gastric contents or media. Palivizumab at 8 μg/mL in pooled gastric contents or virus medium neutralized RSV, but RSV infected Hep-2 cells when palivizumab was at 4 or 2 μg/mL (Figure 6A-E, K–P).

With the flow cytometry analysis of RSV infection with and without palivizumab-containing samples, fluorescence was 1.5-fold lower in medium with palivizumab (6.25 μg/mL) and with RSV-GFP (1.1 × 10^5^ FFA/mL) than in medium with only RSV (no palivizumab) (Figure 7A). However, fluorescence in medium without RSV was 2-fold lower than in medium with RSV and palivizumab, indicating that palivizumab did not completely neutralize RSV. Fluorescence was at the highest level after three days of infection (Figure 7B). A higher concentration of RSV-GFP was needed with flow cytometry compared with the microcopy measurements to obtain a significant difference between positive (infected cells) and negative (non-infected) cells. In the presence of RSV-GFP, fluorescence was lower in human milk or medium with palivizumab (25, 50 or 100 μg/mL) than in milk or medium without palivizumab (Figure 7C). GFP-expressed cells were lower in human milk or medium with 100 μg/mL of palivizumab compared with milk or medium with 25 μg/mL of palivizumab (Figure 7C). Fluorescence did not differ between human milk and virus medium in the presence or absence of palivizumab (Figure 7C). No infected cells were detected in human milk or medium with 100 μg/mL of palivizumab whereas an increase of GFP-expressed cells was observed with a reduction of palivizumab concentration (Figure 7D).

## 4. Discussion

RSV is the most common cause of lower respiratory tract infection that leads to hospitalization and mortality in former preterm infants. Human milk feeding offers protection against respiratory and gastrointestinal tract infections during infancy while present in the diet. The mechanisms by which human milk offers immunity are not fully understood. Human milk immunoglobulins, which are predominantly SIgA and in smaller amounts IgA, IgM, and IgG isotypes [5,6], play a role in providing passive immunity [15], particularly during the first few weeks of life when the infant gastrointestinal tract has yet to secrete significant amounts of endogenous SIgA on the surface of the intestinal mucosa. The repertoire of specificity of milk antibodies is shaped by the history of antigen exposure in the mother, including via the respiratory and digestive systems [16]. Whether RSV-specific IgG from human milk play a role in preventing RSV infection when ingested by infants is unknown. As among antibody isotypes IgG has the highest opsonization activity, if it were to be absorbed across the gut into the bloodstream, it could activate the adaptive immune response to protect against infection in the respiratory tract. However, whether oral IgG is absorbed intact in human infants remains unknown. 

If absorption were to occur, the development of oral recombinant RSV-specific antibodies could be an appealing approach to prevent pneumonia or bronchiolitis in preterm infants. To function optimally, these antibodies would need to survive digestion to the point of absorption. Orally administered recombinant antibodies against enteric pathogens would also need to survive digestion to prevent pathogen infection throughout the gut. In this study, palivizumab was used as model recombinant antibody to examine the effects of simulated infant digestion on binding capacity. The neutralizing capacity of palivizumab against RSV was also evaluated in human milk, gastric contents and stool samples from preterm infants.

Palivizumab is a humanized (95% human and 5% mouse sequences) monoclonal antibody (IgG1) that is specific for the RSV surface fusion protein (F-protein) [17]. Binding of the RSV F-protein to the host lung epithelial cell receptor is required for infection, and palivizumab’s binding to the F-protein blocks the fusion of RSV on the lung epithelial cell. At minimum, palivizumab’s idiotype region must retain enough structure to bind the F-protein and block infection. We examined several conditions that could affect palivizumab stability in the setting of modeling recombinant IgG1 function in the gastrointestinal tract. We used a palivizumab-specific anti-idiotype antibody to evaluate the binding capacity of palivizumab IgG1 in human milk and digested milk because we did not want to assay the binding activity of endogenous human milk anti-RSV F protein IgG.

Simulated digestive fluid is a complicated mixture, the components of which may non-specifically inhibit binding in the assay. The remaining binding capacity of palivizumab in fresh, non-treated palivizumab spiked into milk, gastric, and intestinal samples was determined via ELISA. During the spiking assay, palivizumab was diluted with blocking buffer with and without each type of sample (milk, gastric and intestinal fluids, and combined milk, gastric and intestinal fluid) to evaluate the effect of the simulated digestive fluid on the binding assay (data not shown). We found a significant reduction of binding capacity when intestinal fluid was used whereas milk or gastric samples did not affect the binding capacity. Therefore, the results of this present study are with the addition of simulated digestive fluid into to the blank in order to subtract the background effect.

A freeze and thaw cycle at –20 °C or –80 °C did not affect palivizumab stability, suggesting that palivizumab can be safely frozen in human milk without being destroyed. This control was important to show that freezing of clinical samples would not degrade palivizumab activity. Palivizumab binding capacity was reduced in human milk after 1 h at 37 °C and was not stable in simulated term infant gastric or intestinal conditions. These results are in accordance with our recent study that demonstrated a degradation of palivizumabs (in IgG-, sIgA- and IgA-formats) during incubation in infant gastric and intestinal fluids (ex vivo digestion) [11]. Moreover, palivizumab stability slightly varied depending of the mother’s milk used, which suggest that there can be variability in milk components across mothers that affect palivizumab stability. These components could be milk enzymes or proteolytic bacteria. It is well accepted that the composition of milk between mothers vary in macronutrients, micronutrients, immune components, and commensal bacteria. Therefore, the initial efficiency of recombinant RSV-specific antibodies (or other recombinant antibodies) in human milk may vary between mothers. One of the three tested human milk samples inhibited palivizumab detection by 40% without any simulated gastric or intestinal digestion, indicating that palivizumab was likely cleaved by endogenous milk enzymes or bacterial proteases. Infants also have different digestive fluid proteolytic activity [5,10] and therefore, a future clinical study is needed to evaluate the survival of palivizumab in different neonatal digestive systems.

The binding capacity of palivizumab decreased in simulated gastric and intestinal fluids both with and without the addition of pepsin and pancreatin. This finding indicates that endogenous milk proteases or native milk bacterial proteases were active in degrading palivizumab in both simulated gastric and intestinal conditions. The reduction of palivizumab binding capacity with the addition of pepsin and pancreatin indicates the extent of their contribution to the overall proteolytic degradation. Bile salts did not affect the reduction of the binding capacity of palivizumab. The optimum storage pH for palivizumab is 6.0 (Synagis^®^ liquid solution); however, adjustment of the reactions to pH to 4, 7, and 8 did not decrease palivizumab stability. The binding and neutralizing capacity of palivizumab in biological samples may differ between term and preterm infants. Our previous study demonstrated that gastric pH in preterm infants (GA 24–32 weeks) was comparable to that in term infants (GA 38–40 weeks) 1–3 weeks of postnatal age (overall mean: pH 4.4 ± 0.2) collected at 2 h post-feeding [10]. However, we observed that protease activity in the gastric contents was lower in preterm infants than in term infants [10]. Therefore, the survival of palivizumab may be higher in preterm infants than in term infants due to their lower digestive capacity. More study is needed to compare the binding and neutralizing capacity of palivizumab in digestive fluids post-feeding between preterm and term infants.

For the first time, we demonstrated that palivizumab in human milk, gastric contents, and stool samples blocked RSV infection of Hep-2 cells. A higher concentration of palivizumab was needed to block the syncytia formation on Hep-2 cells by RSV in gastric contents or virus medium (observed with the inverted microscope) than in human milk or stool sample, suggesting either that palivizumab has a higher neutralizing capacity when incubated in human milk or infant stool or that these matrices have a higher background neutralization capacity. The reduction of palivizumab concentration in human milk and virus medium increased the fluorescence detected by flow cytometry for GFP-expressing cells due to RSV-GFP infection, indicating a dose response of palivizumab to prevent the infection of cells. GFP-expressing cells did not differ between human milk or medium at different concentrations of palivizumab. The differences in results for the effect of the matrices between flow cytometry and microscopy are partially due to the difference of RSV-GFP concentration added in the mixture during the neutralization step, where flow cytometry conditions required a higher level of RSV-GFP to detect a difference in fluorescence between infected cells and non-infected cells. The neutralizing capacity of palivizumab was performed in diluted supernatant of biological samples to reduce the antiviral effect of matrices on Hep-2 cells or on RSV. The neutralization may differ between undiluted and diluted biological samples. 

Our recent study demonstrated that naturally-occurring human milk polyclonal IgG was stable in the preterm infant stomach but was partially degraded in the term infant stomach [5]. Preterm infants fed pasteurized human milk supplemented with substantial amounts of serum-derived IgA and IgG excreted a measurable amount of immunoglobulins in feces [8], indicating that when supplemented in large amounts, orally-supplemented Ig can survive to the stool. In another study, IgG was recovered in the ileum of adult humans fed a bovine colostrum Ig concentrate [18]. Taken together, these studies indicate that at least some polyclonal Ig can survive transit through the human digestive system, aided by sheer amounts given, post-translational modifications that limit proteolysis, or other factors.

A limitation of this study was the small number of mother’s milk samples used to detect variability between mothers. At minimum, we did find variability between the three mother’s milk samples used in this study.

This study revealed that palivizumab was partially digested during simulated term infant digestion. Palivizumab degradation likely resulted from both native milk proteases and added gastrointestinal proteases. This investigation indicates that human milk differs between mothers in degradation activity that affect antibody survival across infant digestion. These factors could be variations in milk proteases between mothers. This study demonstrated a higher neutralizing capacity of palivizumab in human milk or infant stool samples than in gastric content or virus medium using microscopy. We also demonstrated a dose response of neutralizing capacity of palivizumab where a reduction of its concentration increased the fluorescence (infection) from GFP-expressed Hep-2 cells in both human milk and virus medium. These results suggest that undigested and digested matrices could change the binding and neutralizing capacity of viral pathogen-specific antibodies. Future studies will examine the in vivo degradation of palivizumab exposed to gastrointestinal fluids obtained in infants administered palivizumab enterally. For optimal application of oral antibodies to prevent infectious diseases, the development of new pathogen-specific recombinant antibodies that are more resistant to digestion is needed.

## Figures and Tables

**Figure 1 nutrients-12-01904-f001:**
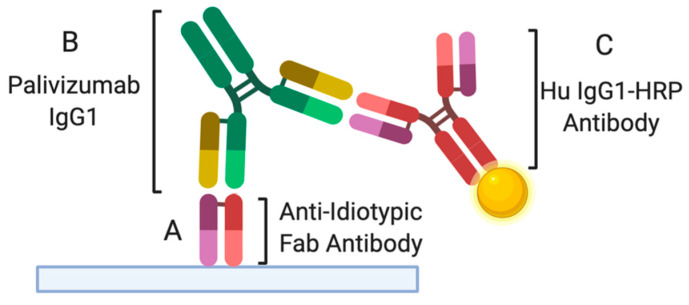
Illustration of palivizumab ELISA. (**A**) Anti-idiotypic Fab antibody is coated in 96-well plate and can specifically recognize (**B**) the monoclonal antibody palivizumab IgG1 in biological samples. (**C**) After washing the biological samples, human anti-palivizumab IgG1 conjugated with horseradish peroxidase (HRP) is added to detect palivizumab.

**Figure 2 nutrients-12-01904-f002:**
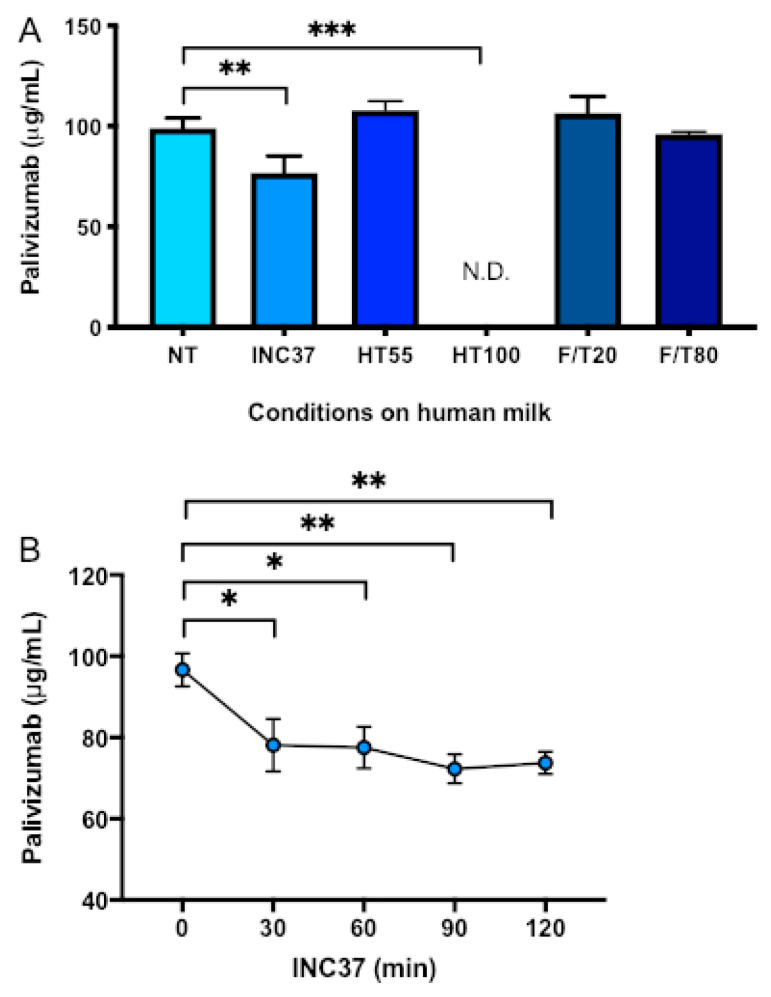
Binding capacity of palivizumab in human milk with different conditions. (**A**) Palivizumab in human milk; non-treated (NT), incubated at 37 °C for 1 h (INC37), heat-treated at 55 °C for 30 min (HT55) and at 100 °C for 30 min (HT100), and frozen and thawed at –20 °C (F/T20) and at –80 °C (F/T80). Values are mean ± SD, *n* = 3. (**B**) Palivizumab in pooled mother’s milk samples incubated at 37 °C for 2 h (INC37) and measured every 30 min. Values are mean ± SD, *n* = 6. Asterisks show statistically significant differences between groups (*** *p* < 0.001; ** *p* < 0.01; * *p* < 0.05) using one-way ANOVA followed by Dunnett’s multiple comparison tests.

**Figure 3 nutrients-12-01904-f003:**
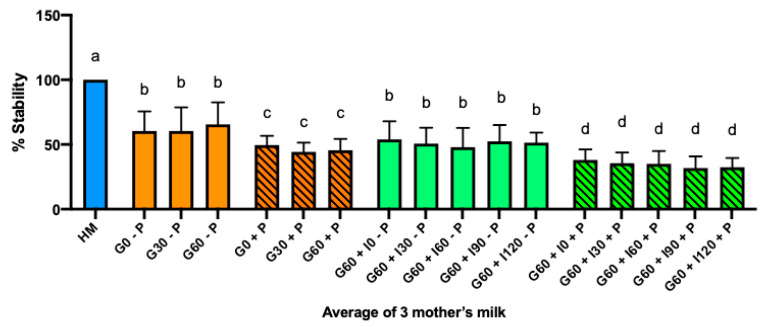
Palivizumab stability across simulated term infant digestion from three different mother’s milk samples. Palivizumab in the average of three mother’s milk (HM) samples; in gastric conditions at 0 min (G0 – P), at 30 min (G30 - P) and at 60 min (G60 – P) without protease (pepsin); in intestinal conditions at 0 min (G60 + I0 – P), at 30 min (G60 + I30 – P), 60 min (G60 + I60 – P), 90 min (G60 + I90 – P) and 120 min (G60 + I120 – P) without protease (pancreatin). Values are mean ± SD from 3 mother’s milk samples with 6 replicates for each condition. Letters a, b, c and d show statistically significant differences between groups (*p* < 0.05) using one-way ANOVA followed by Tukey’s multiple comparison tests.

**Figure 4 nutrients-12-01904-f004:**
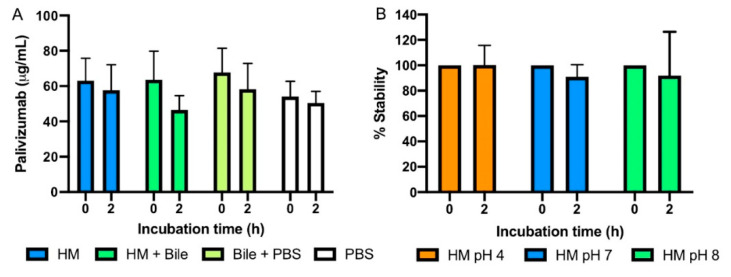
Effect of bile salts and pH on palivizumab stability. (**A**) Effect of bile salts on palivizumab stability at 0 h and after incubation (2 h) at 37 °C with shaking at 300 rpm in presence or in absence of bile salts and/or milk. HM, milk with no treatment; HM + bile, milk and bile; Bile + PBS, bile in PBS; and PBS, PBS alone. Values are mean ± SD from pooled mother’s milk samples with 3 replicates at each of 2 dilutions. Two-way ANOVA followed by Tukey’s multiple comparisons test where means were compared between time 0 and 2 h for each sample. (**B**) Effect of pH on palivizumab stability at 0 h and 2 h at 37 °C with shaking at 300 rpm with pH adjusted to 4, 7, and 8. HM pH 4, milk adjusted pH to 4; HM pH 7, milk adjusted pH to 7; and HM pH 8, milk adjusted pH to 8. Values are mean ± SD from 3 mother’s milk samples with 3 replicates at each of 2 dilutions. Two-way ANOVA followed by Tukey’s multiple comparisons test where means were compared between time 0 and 2 h for each sample.

**Figure 5 nutrients-12-01904-f005:**
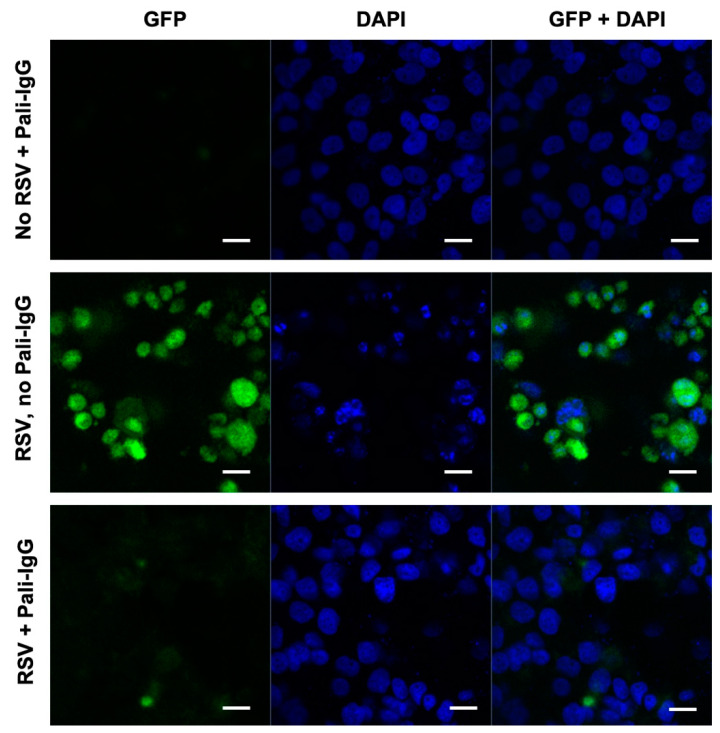
Neutralizing palivizumab IgG against the human respiratory syncytial virus with a green fluorescent protein (RSV-GFP) in pooled human milk using confocal microscopy. RSV-GFP suspension containing 50% tissue culture infectious dose (TCID50) (3.4 × 10^4^ focus forming units (FFU)/mL) was added to pooled human milk (filtered supernatant diluted in 1:10) with 0 and 8 μg/mL of palivizumab for 2 h at 37 °C. The mixtures were transferred to confluent Hep-2 cells for 2 h at 37 °C. A blue fluorescent nucleic acid stain, 4,6-diamidino-2-phynylindole dilactate (DAPI) was added to perform an immunofluorescence staining of Hep-2. GFP expression in infected cells was monitored using a confocal microscope system (Zeiss LSM 780 NLO, White Plains, NY, USA) at 64× objective. Scale bar = 100 μm).

**Figure 6 nutrients-12-01904-f006:**
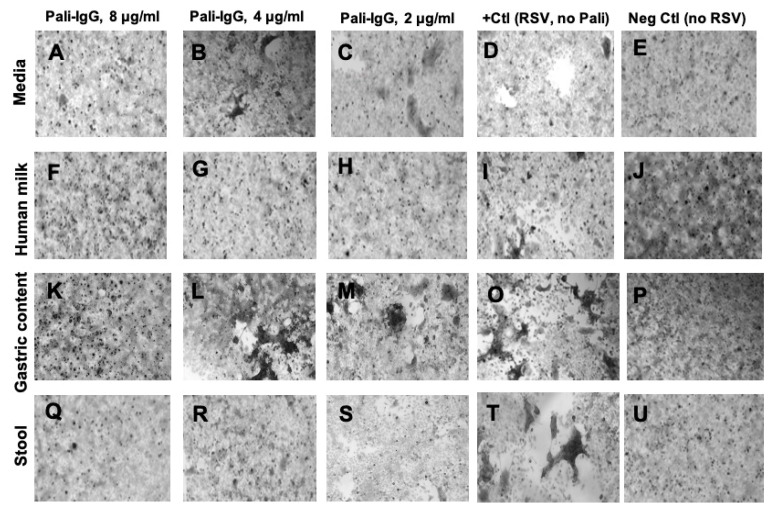
Neutralizing capacity of palivizumab IgG against the human respiratory syncytial virus with a green fluorescent protein (RSV-GFP) in pooled human milk, pooled gastric contents and pooled stool from preterm infants using inverted microscopy. RSV-GFP suspension containing 50% tissue culture infectious dose (TCID50: 3.41 × 10^4^ focus forming units (FFU)/mL) was added in virus medium and filtered supernatant of diluted pooled human milk (1:10), pooled gastric contents (1:20) or pooled stool samples (0.2 g/mL and 1:25) with 0, 2, 4, and 8 μg/mL of palivizumab for 2 h at 37 °C, 5% CO_2_. The mixtures were transferred to confluent Hep-2 and incubated for 2 h at 37 °C. The neutralizing capacity of palivizumab in each condition was monitoring using an Olympus IX51 fitted with a Q Image camera (20× objective) to compare the cytopathic effects. Palivizumab at (**A**) 8 μg/mL, (**B**) 4 μg/mL, (**C**) 2 μg/mL, (**D**) 0 μg/mL in virus medium with RSV and (**E**) at 8 ug/mL without RSV in virus medium. Palivizumab at (**F**) 8 μg/mL, (**G**) 4 μg/mL, (**H**) 2 μg/mL, (**I**) 0 μg/mL in pooled human milk with RSV, and (**J**) at 8 ug/mL without RSV in pooled human milk. Palivizumab at (K) 8 μg/mL, (**L**) 4 μg/mL, (**M**) 2 μg/mL, (**O**) 0 μg/mL in pooled gastric contents from preterm infants with RSV and (**P**) at 8 ug/mL without RSV. Palivizumab at (**Q**) 8 μg/mL, (**R**) 4 μg/mL, (**S**) 2 μg/mL, (**T**) 0 μg/mL in stools from preterm infants with RSV and (**U**) at 8 ug/mL without RSV.

**Figure 7 nutrients-12-01904-f007:**
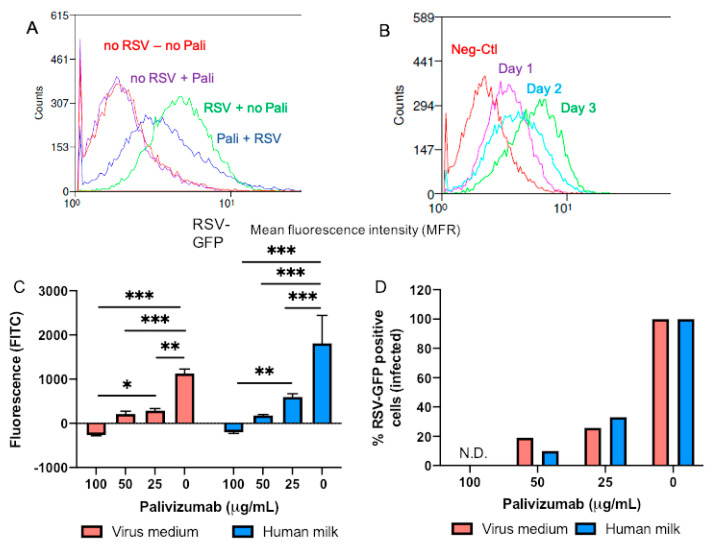
Neutralizing capacity of palivizumab IgG against the human respiratory syncytial virus with a green fluorescent protein (RSV-GFP) in pooled human milk and virus medium using flow cytometry. RSV-GFP suspension (1.1 × 10^5^ focus forming units (FFA)/mL) was added in virus medium and filtered supernatant of diluted pooled human milk (1:10) with 0, 25, 50, and 100 μg/mL of palivizumab for 2 h at 37 °C, 5% CO_2_. The mixtures were transferred to confluent Hep-2 and incubated for 2 h at 37 °C. Fluorescence measured by flow cytometry (FACSCalibur) in Hep-2 cells incubated with RSV-GFP (1.1 × 105 FFA/mL) (**A**) in virus medium with 0 and 100 μg/mL of palivizumab after 3 days of infection; and (**B**) at day 1, day 2, and day 3 of infection in the absence of palivizumab (neg Ctl: no RSV and no palivizumab). (**C**) Fluorescence measured by flow cytometry in the presence of RSV-GFP in human milk or virus medium with palivizumab (0, 25, 50 and 100 μg/mL). *** *p* < 0.001; ** *p* < 0.01; * *p* < 0.05. (**D**) Percentage of GFP-expressed Hep-2 cells (or infected cells) in the presence of RSV-GFP in human milk or virus medium with palivizumab.

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
