# Peer review of "Binding and Neutralizing Capacity of Respiratory Syncytial Virus (RSV)-Specific Recombinant IgG Against RSV in Human Milk, Gastric and Intestinal Fluids from Infants"

_nutrients, 2020, doi:10.3390/nu12071904_

Round 1
Reviewer 1 Report
This manuscript is clearly written. It compared stability of palivizumab under various in vitro gastrointestinal digestions. The authors used an assumed protein concentration of breast milk to perform the in vitro digestion experiments and then used ELISA to evaluate integrity of palivizumab after digestion. The three breast milk samples may have significantly different protein concentrations, but the same concentration of proteases was used. Results generated by this procedure are not reliable and cannot be compared between samples, and no conclusion can be drawn. In addition, ELISA is not an appropriate method to evaluated protein stability/integrity after gastrointestinal digestion.
- Line 18, remaining activity; ELISA was used to test the integrity of palivizumab not the activity in the current study, but the authors used activity to describe the ELISA results in the manuscript. ELISA is not an appropriate approach to assess integrity of digested proteins since fragments containing epitopes can be detected by ELISA.
- Lines 45-47, since no study has demonstrated the absorption of human milk IgG in the human gut and then to the bloodstream, it is possible that milk IgG cannot enter the circulation system. What is the basis of the current study?
- Lines 48-49, Is the % of IgG, IgA, and IgM % of the total milk protein or the total immunoglobulin?
- How were the milk samples collected? Were the concentrations of milk samples measured?
- Line 106, why were the three pH values chosen?
- Line 122, assumption that human milk is 10 mg/mL protein; why wasn’t the concentration of breastmilk samples measured? The protein concentration of breastmilk is affected by various factors. It is most likely that the concentrations of the three breast milk samples are different. However, the authors used the same concentration of enzymes for the three samples, which cannot generate reliable results.
- Line 175, how was the antigen-binding activity measured?
- Figure1&2, It is hard to obtain information when only one x-axis is labeled.PE and PA may represent pepsin and pancreatin, respectively.
Minor:
- Line 20, “a simulated gastric and jejunal fluids” needs to be corrected
- Line 111, gastrointestinal fluids were not used in the current study.
Author Response
This manuscript is clearly written. It compared stability of palivizumab under various in vitrogastrointestinal digestions. The authors used an assumed protein concentration of breast milk to perform the in vitro digestion experiments and then used ELISA to evaluate integrity of palivizumab after digestion. The three breast milk samples may have significantly different protein concentrations, but the same concentration of proteases was used.
>> Thank you. We added the same concentration of recombinant IgG (100 ug/mL) in the 3 mother’s milk samples. However, one mothers (Figure 3B) had likely higher protease activity in her milk, thus palivizumab IgG was digested during the first step in human milk. It is why we have a lower concentration in human milk for mother’s milk #2.
Results generated by this procedure are not reliable and cannot be compared between samples, and no conclusion can be drawn. In addition, ELISA is not an appropriate method to evaluated protein stability/integrity after gastrointestinal digestion.
>> We are not evaluating protein stability/integrity but the binding capacity of palivizumab, a recombinant antibody IgG, in human milk and digested milk. ELISA is an appropriate method to evaluate the binding capacity of antibody IgG in biological samples. We added more information to explain the palivizumab ELISA (Figure 1 and line 177-180).
- Line 18, remaining activity; ELISA was used to test the integrity of palivizumab not the activity in the current study, but the authors used activity to describe the ELISA results in the manuscript. ELISA is not an appropriate approach to assess integrity of digested proteins since fragments containing epitopes can be detected by ELISA.
>> We evaluated the remaining binding capacity of palivizumab IgG using ELISA. If the epitope of IgG is destroyed or cleaved, the anti-idiotypic Fab antibody coated on the microplate surface cannot bind to palivizumab IgG. If palivizumab is not attached to this anti-idiotypic antibody, the washing step will remove the unbinding palivizumab and thus, we will not detect it at the final step. We added the Figure 1 to illustrate the ELISA used to detect the binding capacity of palivizumab.
- Lines 45-47, since no study has demonstrated the absorption of human milk IgG in the human gut and then to the bloodstream, it is possible that milk IgG cannot enter the circulation system. What is the basis of the current study?
>> It is unknown whether milk IgG can enter the circulation system. Our future study will determine whether palivizumab IgG can enter to the circulation system after the absorption in the human gut. This information was on lines 52-53.
- Lines 48-49, Is the % of IgG, IgA, and IgM % of the total milk protein or the total immunoglobulin?
>>This is total immunoglobulin for IgA, IgM and IgG. We did not evaluate the total protein concentration in this study.
- How were the milk samples collected? Were the concentrations of milk samples measured?
>> Milk samples were collected at home with clean electric breast pumps into sterile plastic containers and stored immediately at –20°C in deep freezers. The breast was cleaned with a wet washcloth (no soap or alcohol) before pumping. We added this information in the method section (line 99-102)
>> The total protein concentration was not measured because we are measuring a recombinant IgG1 (palivizumab).
- Line 106, why were the three pH values chosen?
>>We selected these values to evaluate 1) an acidic condition (pH4) like in infant gastric fluid, 2) a neutral condition (pH7) like in human milk and 3) an alkaline condition (pH8) like in duodenal fluid. We added this information (lines 120-121)
- Line 122, assumption that human milk is 10 mg/mL protein; why wasn’t the concentration of breastmilk samples measured? The protein concentration of breastmilk is affected by various factors. It is most likely that the concentrations of the three breast milk samples are different. However, the authors used the same concentration of enzymes for the three samples, which cannot generate reliable results.
>>Because we used palivizumab as a supplement of antibody, the concentration of total protein had no effect on our results.
- Line 175, how was the antigen-binding activity measured?
>> We changed this word for binding capacity. Binding capacity of palivizumab was measured with ELISA (line 265).
- Figure1&2, It is hard to obtain information when only one x-axis is labeled.PE and PA may represent pepsin and pancreatin, respectively.
>> We identified “+ P” for samples with protease and “- P” for samples without protease. As the figure is already surcharged of information, this annotation allowed us to simplify and help the reader to understand the Figure 3 and 4.
Minor:
- Line 20, “a simulated gastric and jejunal fluids” needs to be corrected
>> We corrected this sentence (“the” and “duodenal”). (line 20).
- Line 111, gastrointestinal fluids were not used in the current study.
>>We specified that gastric and duodenal fluids were in vitro (line 126).
Reviewer 2 Report
A study entitled "Proteases contribution differently between mother's milk to the digestion of viral pathogen-specific recombinant IgG in simulated gastric and intestinal fluids from infant" aimed to investigate the palivizumab stability and binding capacity in human milk which has been exposed to in vitro simulated gastric and intestinal digestion.
The study is well conduct, although the number of milk donors is small (only 3). Therefore strong conclusions could not be made. One of the major concern is protein stability at different pH.
Have the authors study the solubility of antibody together with milk proteins at pH4? It is known that some of the milk proteins could precipitate together with white micelles which can drag down the spiked palivizumab. Another point is that authors see differences between mothers. What was the protein amount of the milk in each donor and if there is a difference between proteins and lipids compositions between donors?
Figure 1 showed different incubations at 37C, 55 and freeze-thawing. My concer here is why authors incubated the antibody at 37 and 55 at different time points? Have they been testing "stability" i.e. the binding capacity of the ab at 30min incubation at 37, or more importantly to at 1h incubation at 55C?
Beside ELISA and measurement of the amount of the palivizumab, have the authors were able to detect the antibody with some other methodology (western-blot, size-exclusion chromatography, etc)? Protein degradation by protease could lead to peptides which could be biologically active and for opsonisation ab should be intact, correct? Therefore it will be interesting to see if binding capacity comes from intact or digested-degraded protein.
Minor
Page 1, abstract, line 18. The aim of was to compare the remaining "binding capacity" instead of activity.... The same in line 22 in the sentence starting "The reduction of palivizumab activity...
Line 23 "likely due to their differences in milk proteases" as mentioned above, amount of other proteins which could aggregate and take down the palivizumab should be considered.
Concluding sentence should be rephrased and be more clear.
Author Response
A study entitled "Proteases contribution differently between mother's milk to the digestion of viral pathogen-specific recombinant IgG in simulated gastric and intestinal fluids from infant" aimed to investigate the palivizumab stability and binding capacity in human milk which has been exposed to in vitro simulated gastric and intestinal digestion.
The study is well conduct, although the number of milk donors is small (only 3). Therefore, strong conclusions could not be made. One of the major concern is protein stability at different pH.
Have the authors study the solubility of antibody together with milk proteins at pH4? It is known that some of the milk proteins could precipitate together with white micelles which can drag down the spiked palivizumab. Another point is that authors see differences between mothers. What was the protein amount of the milk in each donor and if there is a difference between proteins and lipids compositions between donors?
>> Thank you. We studied the binding capacity of palivizumab in human milk at pH 4. We observed some precipitated milk proteins, but the pH 4 did not affect the binding capacity/stability of palivizumab. We did not measure the total protein or lipid concentration in human milk because we are studying a non-milk protein (recombinant palivizumab IgG1).
Figure 1 showed different incubations at 37C, 55 and freeze-thawing. My concern here is why authors incubated the antibody at 37 and 55 at different time points? Have they been testing "stability" i.e. the binding capacity of the ab at 30min incubation at 37, or more importantly to at 1h incubation at 55C?
>> Thank you for your comment. We added our results for the incubation of palivizumab in human milk for 2 h where we measured the binding capacity at each 30 min (see Figure 1B). We tested 55C for 30 min as this heat-treatment is generally used in immunology/microbiology to inactivate viruses in media.
Beside ELISA and measurement of the amount of the palivizumab, have the authors were able to detect the antibody with some other methodology (western-blot, size-exclusion chromatography, etc)? Protein degradation by protease could lead to peptides which could be biologically active and for opsonisation ab should be intact, correct? Therefore, it will be interesting to see if binding capacity comes from intact or digested-degraded protein.
>>Thank you for this great comment. We added supplemental results for neutralizing capacity of palivizumab against RSV-GFP using confocal and inverted microscope as well as flow cytometry. Moreover, palivizumab ELISA uses human anti-palivizumab antibodies that are paratope specific, high affinity, anti-idiotypic antibody that specifically recognizes the monoclonal palivizumab IgG in biological samples.
Minor
Page 1, abstract, line 18. The aim of was to compare the remaining "binding capacity" instead of activity.... The same in line 22 in the sentence starting "The reduction of palivizumab activity...
>> We corrected for “binding capacity” everywhere in the manuscript.
Line 23 "likely due to their differences in milk proteases" as mentioned above, amount of other proteins which could aggregate and take down the palivizumab should be considered.
>> This sentence is referring to palivizumab in human milk (neutral pH), therefore the milk proteins were intact and should not take down the palivizumab.
Concluding sentence should be rephrased and be more clear.
>> We changed the concluding sentence to be clearer (lines 349-351).
Reviewer 3 Report
The concept behind the research and the manuscript are of high value to the field, and I commend the authors for this endeavor; however, I feel there are simple experiments that could fill in some of the holes in the current draft. Namely, the author's major take away is the reduction of Palivizumab concentration after gastric and intestinal digestion. Although this may be interesting from the perspective of antibody 'survivability' through the infant's gastrointestinal tract, it does not provide clinically meaningful data for which the authors' suggest '...digestion may decrease palivizumab's RSV neutralizing capacity." in the discussion section. It would behoove the author's to complete experiments beyond sandwich ELISA that show synthetic digestion through the infant's gastrointestinal tract affects Palivizumab's ability to bind RSV domains. This could be completed using direct ELISA experiments and would strengthen the conclusions put forth by the authors.
Additional minor points include providing additional information in the introduction identifying why the authors chose the specific pHs tested throughout the paper. It has been reported that term infants have a gastric pH 2-5 (dependent on postnatal age), while preterm infants have a gastric pH >7.
Finally, it was not clear throughout the paper whether these findings are applicable to a preterm or term infant population. Of note, the authors describe that RSV infection, '...can induce serious infection in preterm infants...' yet complete the experiments with term breast milk. The authors should note the difference in protein content, etc in preterm versus term infants. Alternatively, these results could be tailored to a term population of which some of the statistics around national RSV infection rates in the introduction come from and gastric pH values used in the experiment related more closely to.
Author Response
The concept behind the research and the manuscript are of high value to the field, and I commend the authors for this endeavor; however, I feel there are simple experiments that could fill in some of the holes in the current draft. Namely, the author's major take away is the reduction of Palivizumab concentration after gastric and intestinal digestion. Although this may be interesting from the perspective of antibody 'survivability' through the infant's gastrointestinal tract, it does not provide clinically meaningful data for which the authors' suggest '...digestion may decrease palivizumab's RSV neutralizing capacity." in the discussion section. It would behoove the author's to complete experiments beyond sandwich ELISA that show synthetic digestion through the infant's gastrointestinal tract affects Palivizumab's ability to bind RSV domains. This could be completed using direct ELISA experiments and would strengthen the conclusions put forth by the authors.
>> Thank you for this comment. We added supplemental results to evaluate the neutralizing capacity of palivizumab in human milk, gastric content and stool samples from preterm infants using a confocal and inverted microscope and flow cytometry. These methods provide clinically meaningful data for the neutralizing capacity of antibodies in human milk and infant biological samples.
Additional minor points include providing additional information in the introduction identifying why the authors chose the specific pHs tested throughout the paper. It has been reported that term infants have a gastric pH 2-5 (dependent on postnatal age), while preterm infants have a gastric pH >7.
>> We added in the method section why selected these specific pHs. pH 4 (simulating the acidic pH in gastric fluid), pH 7 (simulating the neutral pH in human milk) or 8 (simulating the alkaline pH in duodenal fluid). (line 119-121). We also added in discussion that the binding and neutralizing capacity of palivizumab in biological samples may differ between term and preterm infants. Term infants have a gastric pH 2–5 (dependent on postnatal age), while preterm infants have often a gastric pH >5. Our previous paper demonstrated that protease activity in the gastric contents was lower in preterm infants than in term infants.
Finally, it was not clear throughout the paper whether these findings are applicable to a preterm or term infant population. Of note, the authors describe that RSV infection, '...can induce serious infection in preterm infants...' yet complete the experiments with term breast milk. The authors should note the difference in protein content, etc in preterm versus term infants. Alternatively, these results could be tailored to a term population of which some of the statistics around national RSV infection rates in the introduction come from and gastric pH values used in the experiment related more closely to.
>>Thank you for this comment. We added supplemental data using biological samples (gastric contents and stool samples from preterm infants. Therefore, these findings are applicable to a preterm or term infant population. Moreover, preterm infants are often fed with pasteurized donor milk (usually from term-delivery mothers). We added in the discussion that more studies are needed to compare the binding and neutralizing capacity of palivizumab in human milk and digested milk between preterm and term infants (line 492-493).
Reviewer 4 Report
The paper is another consideration of human breast milk by a group with a strong publication record in this area. The paper is well written and appropriately referenced, with the data supporting the conclusions presented by the authors. I do have several concerns with the manuscript in the present form:
- There is no statement of IRB approval for use of human milk samples. The samples were donated from the mother's milk bank, but I still think use of human tissues or specimens should warrant IRB approval, as I'm sure the mom's did not intend their donated milk to be used for experiments. The IRB would probably grant a waiver but I think IRB review is still critical.
- The overall nature of this investigation is somewhat contrived, especially considering the authors have recently published a much more relevant paper looking at degradation of naturally occurring breastmilk antibodies to RSV. Synthetic anti-RSV is not given enterally, and is not produced for this purpose, so to then test how this IM antibody vaccine stands up to gastric contents seems not that important. The argument is that it gives information about antibodies in the BM in general, but I think the paper by Lueangsakulthai they have already published is much more clinically relevant on that end. I'm not sure how much this report adds.
- More specific concerns include:
- Figure 1 and line 175 - text states that ag binding activity was reduced 22.5% but graph shows antibody concentration on y-axis. These are not the same measures.
- Why were only 3 mother's milk samples used? Seems like a small n-number for this study where samples were donated and perhaps not that hard to come by.
- Figure 2 – I don’t understand the rationale for including all of the sepsrate mother’s info here. It is then combined and presented again in parts A and B of figure 3. Very redundant. The statistics and SD from figure 3 tell enough info that figure 2 seems unnecessary.
- Figure 2 & 3 – the use of a, b, c, d to designate differences between groups does not allow the reader to understand which groups are different. More information is needed to clarify.
- Figure 3 – stability is now used to standardize the levels for differences in baseline. This could have been done from the outset. Unclear why authors are displaying the same data in so many different ways.
- Fig 3C – why is there a line graph here? The samples are independent and not related, line graph seems inappropriate, would stick with the bar graph used for all the other depictions of the same data in the earlier figures.
- Fig 4A – why were samples pooled for this analysis and not for others?
- Wondering why an anti-idiotype antibody was used to detect RSV when there is an F-protein binding ELISA available. In the discussion you point out that F protein binding domain needs remain intact after digestion for palivizumab antibody activity, which caused me to wonder why not assess this in the paper since there are tools available to do that.
Author Response
The paper is another consideration of human breast milk by a group with a strong publication record in this area. The paper is well written and appropriately referenced, with the data supporting the conclusions presented by the authors. I do have several concerns with the manuscript in the present form:
- There is no statement of IRB approval for use of human milk samples. The samples were donated from the mother's milk bank, but I still think use of human tissues or specimens should warrant IRB approval, as I'm sure the mom's did not intend their donated milk to be used for experiments. The IRB would probably grant a waiver but I think IRB review is still critical.
>> Mothers who donated to the mother’s milk bank signed a consent to use their milk for research. This study was approved by the Institutional Review Board of the Oregon State University. We added this information (line 97-100).
- The overall nature of this investigation is somewhat contrived, especially considering the authors have recently published a much more relevant paper looking at degradation of naturally occurring breastmilk antibodies to RSV. Synthetic anti-RSV is not given enterally, and is not produced for this purpose, so to then test how this IM antibody vaccine stands up to gastric contents seems not that important. The argument is that it gives information about antibodies in the BM in general, but I think the paper by Lueangsakulthai they have already published is much more clinically relevant on that end. I'm not sure how much this report adds.
>> Thank you for this comment. We added supplemental results on the neutralizing capacity of palivizumab in human milk, gastric content and stool samples from preterm infants using confocal and inverted microscope and flow cytometry. This investigation also reported for the first time that mothers can have different protease activities in their milk that can result to different loss of binding capacity antibody in the infant gut. This in vitro model detected these differences between mothers partially due to the absence of variability generally induced when gastric and duodenal fluids from infants are used.
- More specific concerns include:
- Figure 1 and line 175 - text states that ag binding activity was reduced 22.5% but graph shows antibody concentration on y-axis. These are not the same measures.
>> We changed “binding activity” for binding capacity.
- Why were only 3 mother's milk samples used? Seems like a small n-number for this study where samples were donated and perhaps not that hard to come by.
>> We used 3 mother’s milk samples to see possible variability between mothers, but the first aim was to determine whether palivizumab in human milk can remain active after the simulated infant digestion. We did not expect to have some differences between mother’s milk. We added to this limitation (number of samples) to the discussion (line 510-512).
- Figure 2 – I don’t understand the rationale for including all of the sepsrate mother’s info here. It is then combined and presented again in parts A and B of figure 3. Very redundant. The statistics and SD from figure 3 tell enough info that figure 2 seems unnecessary.
>> The figure 2 (figure 3 now) allows the reader to see differences in individual mothers between each step of the simulated digestion. The figure 3 (now figure 4) allows the reader to see the remaining binding capacity of palivizumab in a pooled pasteurized mother’s milk.
- Figure 2 & 3 – the use of a, b, c, d to designate differences between groups does not allow the reader to understand which groups are different. More information is needed to clarify.
>> We added a statistical analysis to determine the difference of palivizumab in human milk between mothers (line 259-260, 286). However, we cannot compare the following steps of the in vitro infant digestion between mothers due to the difference of palivizumab binding activity in mother’s milk 2 compared with mother’s milk 1 and 3.
- Figure 3 – stability is now used to standardize the levels for differences in baseline. This could have been done from the outset. Unclear why authors are displaying the same data in so many different ways.
>> Thank you. We removed two of these figures to reduce the display of the same data.
- Fig 3C – why is there a line graph here? The samples are independent and not related, line graph seems inappropriate, would stick with the bar graph used for all the other depictions of the same data in the earlier figures.
>> We removed the line graph and changed for bar graph.
- Fig 4A – why were samples pooled for this analysis and not for others?
>> We added the results with a pooled pasteurized human milk.
- Wondering why an anti-idiotype antibody was used to detect RSV when there is an F-protein binding ELISA available. In the discussion you point out that F protein binding domain needs remain intact after digestion for palivizumab antibody activity, which caused me to wonder why not assess this in the paper since there are tools available to do that.
>> We selected anti-idiotype antibody to detect palivizumab IgG1 in biological samples because we did not want to measure human milk IgG specific to F protein in this study. We added this information in the discussion (455-457).
Round 2
Reviewer 3 Report
The authors did a good job addressing the concerns around neutralizing antibody effect, which significantly improves the quality of the manuscript and permits the claims made in the discussion.
Author Response
The authors did a good job addressing the concerns around neutralizing antibody effect, which significantly improves the quality of the manuscript and permits the claims made in the discussion.
>> Thank you for your great comments.
Reviewer 4 Report
Thank you for the revisions. this is a much stronger version of the paper and the addition of the neutralization data at the end of the paper really strengthened the relevance for me, thinking of this as a model system for oral antibiotics and passive transfer of maternal IgG. Many of my previous comments were thoroughly addresses with the newer version.
I am happy with the streamlining of the original data into fewer figures, however there remains some redundancy though in fig 3 and 4 as well as Fig 7 and 8.
The use of letters to denote significance in the figures is still not well explained. "Letters a and b show statistically significant differences between time-points." Between which points?? any thing marked with a is different than b?? It's still not clear to me.
Otherwise I think this is a vastly improved manuscript.
Author Response
Thank you for the revisions. this is a much stronger version of the paper and the addition of the neutralization data at the end of the paper really strengthened the relevance for me, thinking of this as a model system for oral antibiotics and passive transfer of maternal IgG. Many of my previous comments were thoroughly addresses with the newer version.
I am happy with the streamlining of the original data into fewer figures, however there remains some redundancy though in fig 3 and 4 as well as Fig 7 and 8.
>> Thank you for your great comments. We changed Figure 3 to Figure S1 to reduce the redundancy. Figure 7 (now Figure 6) and Figure 8 (now Figure 7) are both important to keep in the manuscript as one represents the results using microcopy while the other represents the results obtained with flow cytometry. However, we removed the Fig. 7C and 7D.
The use of letters to denote significance in the figures is still not well explained. "Letters a and b show statistically significant differences between time-points." Between which points?? anything marked with a is different than b?? It's still not clear to me.
>> We changed the letters for asterisks.
Otherwise, I think this is a vastly improved manuscript.